# Antiviral Efficacy of RNase H-Dependent Gapmer Antisense Oligonucleotides against Japanese Encephalitis Virus

**DOI:** 10.3390/ijms241914846

**Published:** 2023-10-02

**Authors:** Shunsuke Okamoto, Yusuke Echigoya, Ayaka Tago, Takao Segawa, Yukita Sato, Takuya Itou

**Affiliations:** 1Laboratory of Preventive Veterinary Medicine and Animal Health, Department of Veterinary Medicine, College of Bioresource Sciences, Nihon University, Fujisawa, Kanagawa 252-0880, Japan; brsh20501@g.nihon-u.ac.jp (S.O.); segawa.takao@nihon-u.ac.jp (T.S.); itou.takuya@nihon-u.ac.jp (T.I.); 2Nihon University Veterinary Research Center, Fujisawa, Kanagawa 252-0880, Japan; bray18116@g.nihon-u.ac.jp (A.T.); sato.yukita@nihon-u.ac.jp (Y.S.); 3Laboratory of Biomedical Science, Department of Veterinary Medicine, College of Bioresource Sciences, Nihon University, Fujisawa, Kanagawa 252-0880, Japan

**Keywords:** Japanese encephalitis virus, gapmer antisense oligonucleotide, locked nucleic acid (LNA), antiviral, RNase H, 3′ UTR stem-loop region, secondary structure, off-target effect, flaviviruses

## Abstract

RNase H-dependent gapmer antisense oligonucleotides (ASOs) are a promising therapeutic approach via sequence-specific binding to and degrading target RNAs. However, the efficacy and mechanism of antiviral gapmer ASOs have remained unclear. Here, we investigated the inhibitory effects of gapmer ASOs containing locked nucleic acids (LNA gapmers) on proliferating a mosquito-borne flavivirus, Japanese encephalitis virus (JEV), with high mortality. We designed several LNA gapmers targeting the 3′ untranslated region of JEV genomic RNAs. In vitro screening by plaque assay using Vero cells revealed that LNA gapmers targeting a stem-loop region effectively inhibit JEV proliferation. Cell-based and RNA cleavage assays using mismatched LNA gapmers exhibited an underlying mechanism where the inhibition of viral production results from JEV RNA degradation by LNA gapmers in a sequence- and modification-dependent manner. Encouragingly, LNA gapmers potently inhibited the proliferation of five JEV strains of predominant genotypes I and III in human neuroblastoma cells without apparent cytotoxicity. Database searching showed a low possibility of off-target binding of our LNA gapmers to human RNAs. The target viral RNA sequence conservation observed here highlighted their broad-spectrum antiviral potential against different JEV genotypes/strains. This work will facilitate the development of an antiviral LNA gapmer therapy for JEV and other flavivirus infections.

## 1. Introduction

Antisense oligonucleotides (ASOs) are short, single-strand nucleic acid analogues having a length of 14–30-mer that bind target RNAs and modulate their functions in a sequence-dependent manner. ASO-based therapeutic approaches have demonstrated potent efficacy in treating various diseases, including viral infections [1,2]. Since the approval of the first ASO drug in eyedrops, fomivirsen, against cytomegalovirus in 1998, nine ASO drugs have been approved only for rare genetic diseases, but no novel ones have yet been approved for viral diseases. However, it is evident that ASOs have great potential for treating viral infections [3], as reported in many clinical and preclinical studies of antiviral ASOs applied to life-threatening viruses, such as influenza A virus [4], Ebola virus, Marburg virus [5], and severe acute respiratory syndrome coronavirus 2 (SARS-CoV-2) [6,7,8,9,10,11] causing the COVID-19 global pandemic. Although there is a need for further investigations towards clinical applications, current studies support the development of viral RNA-targeted ASOs for a therapeutic modality [12].

ASOs primarily function through two mechanisms: RNase H-mediated cleavage/degradation of or steric blocking of target RNAs [13]. ASOs with the former function are called gapmer ASOs, composed of a central DNA segment flanked by short nucleic acid analogue wings. Gapmer ASOs bind to target RNAs and form the DNA/RNA hybrid duplexes via sequence complementarity, allowing for the recruitment of RNase H and the target RNA degradation in the nucleus and cytoplasm [14]. Several gapmer ASOs have received regulatory approval and are under clinical trials for hereditary diseases [15,16]. For viral infections, although the antiviral activity and mechanisms of steric block ASOs, which bind and alter functions of target RNAs, are shown in many virus species and have advanced into clinical trials as described above [4,5,17], those of gapmer ASOs targeting viral RNAs remain unclear. 

Given the ASO property, antiviral gapmer ASOs have the potential advantages of eliminating viral RNAs from infected cells and being flexibly synthesized into mutant viral RNA sequences. Currently, gapmer ASOs modified with locked nucleic acids (LNA gapmers) hold great promise for a new class of antiviral therapeutics to induce RNase H-mediated viral RNA cleavage (Figure 1a) [3,18]. LNAs contain a locked furanose ring by an O2′-C4′-methylene linkage, with which gapmer ASOs are given increased binding affinity to target RNA and nuclease resistance (Figure 1b) [19]. The therapeutic potential of LNA gapmers is demonstrated mainly for various genetic disorders [20,21,22]. Recent studies indicate the efficacy of viral RNA-targeted LNA gapmers against SARS-CoV-2 in vitro and in vivo [6,7,8,9], strongly supporting the possibility of LNA gapmer therapy as an effective strategy for pandemic viruses. However, the efficacy and mechanism of viral RNA-targeted LNA gapmers are poorly understood for viral pathogens involving flaviviruses, which pose significant threats to global public health. In the present study, we examined the antiviral effects of RNase H-dependent LNA gapmers on the Japanese encephalitis virus (JEV) as a model of flaviviruses.

JEV is a mosquito-borne arbovirus that belongs to the genus *Flavivirus* in the family *Flaviviridae*, including other clinically significant flaviviruses such as dengue virus (DNV), yellow fever virus (YFV), West Nile virus (WNV), and zika virus (ZKV) [23,24]. JEV has become the most clinically significant and common causative agent of viral encephalitis in Asia and northern and southeastern Australia [24]. Due to spreading mosquito habitats resulting from global warming and the movement of people, over 3 billion people are at risk of JEV infection [23,25]. Although some vaccination programs are available [26], it is estimated that approximately 69,000 cases occur annually worldwide [23]. In the development of acute encephalitis syndromes, approximately 18% of patients die and approximately 44% experience permanent neurological sequelae that affect the quality of life [23]. Currently, there is no specific treatment for Japanese encephalitis [27].

JEV contains a single-stranded, positive-sense RNA genome nearly 11 kb in length, consisting of a single open reading frame (ORF) and two untranslated regions (UTRs) [23]. The JEV UTRs have been reported to be associated with the specificity of the genotypes (GI to GV), holding the general structures and cis-acting elements of flaviviruses [28]. The conserved sequence (CS) motifs in 5′ and 3′ UTRs are essential for JEV replication. Targeting the CS motifs, studies have shown the efficacy of steric block ASOs using nucleic acid analogues such as peptide-conjugated phosphorodiamidate morpholino oligomer (PMO) [29], vivo-PMO [30], and peptide nucleic acid (PNA) [31]. The study of PNA-modified steric block ASOs also reports stem-loop structures in 5′ and 3′ UTRs as candidate targets for antiviral activity [31]. As such, JEV has a proven record of serving as a tool suitable for pursuing the possibility of viral RNA-targeted therapeutics. However, the ability of antiviral LNA gapmers to regulate JEV infection has yet to be examined.

In this study, in vitro, we demonstrate the efficacy of antiviral LNA gapmer targeting JEV RNA and its action mechanism. We designed several LNA gapmers targeting JEV 3′ UTR and revealed that LNA gapmers targeting the 3′ UTR stem-loop structure are efficacious in inhibiting JEV proliferation in Vero cells. Assays using mismatched LNA gapmers uncovered an antiviral mechanism of LNA gapmers in which their inhibitory effect on JEV proliferation in cells is induced by the RNase H-dependent JEV RNA degradation in a sequence- and LNA-modification-specific manner. Importantly, our LNA gapmers exhibited significant antiviral activity against JEV in a human neuroblastoma cell line of a JEV infection model without apparent cytotoxicity at tested concentrations. An in silico analysis using the human genome database indicated a low possibility of sequence-specific off-target binding of JEV RNA-targeted LNA gapmers to human RNAs. A virus genome database analysis showed high conservation of JEV RNA sequence complementary to our LNA gapmer sequences across most strains composed of genotypes I, III, IV and V, revealing the broad-spectrum potential of the LNA gapmers. This potential applicability was empirically validated in JEV wild-type strains belonging to genotypes I and III, which are the most prevalent in Asia. This work represents the first step toward developing viral RNA-targeted LNA gapmers for treating JEV and flavivirus infections. 

## 2. Results

### 2.1. Design of LNA Gapmers Targeting 3′ UTR of JEV Genomic RNA

Referring to the previous reports regarding the viral genome functions and ASO tests of JEV [29,30,31], we designed nine LNA gapmers composed of 15- to 17-mer on 3′ UTR, an essential region for JEV replication (Table 1) [28]. All tested LNA gapmers were synthesized with phosphorothioate backbones that provide increased nuclease resistance and binding affinity to target RNAs (Figure 1b). For the design, an RNA secondary structure of the partial 3′ UTR (278 nt) of the JaGAr 01 strain was predicted using the RNAfold program (Figure 1c). LNA gapmer 1 was designed by QIAGEN (LG00204874, Venlo, The Netherlands), which targeted the two different regions with the same sequence in 3′ UTR. LNA gapmers 2 and 3 were designed to target the 3′ UTR CS I motif needed for genome cyclization essential for replicating JEV RNA [32]. LNA gapmers 4, 5, 6, 7, 8, and 9 targeted the 3′ UTR stem-loop region so that LNA gapmers can effectively bind with the single-stranded structure formed by the region. The length or proportion of LNA and DNA of the stem-loop-targeted LNA gapmers was altered to examine how such modifications affect binding affinity to target RNA and degradation efficiency. Four controls of LNA gapmers, designed not to bind JEV and human RNAs in theory, were prepared to eliminate the potential false positive results.

### 2.2. Inhibitory Effect of LNA Gapmers on JEV Proliferation in Vero Cells

To find effective target regions and sequences for inhibiting JEV proliferation, we screened the designed LNA gapmers in Vero cells infected with the JEV JaGAr 01 strain at 0.1 multiplicity of infection (MOI), which belongs to genotype III and is one of the proven standard strains in JEV studies [33]. The results showed that JEV RNA-targeted LNA gapmers can inhibit JEV proliferation, as represented by plaque assay (Figure 2a and Appendix A). In particular, the stem-loop-targeted LNA gapmers 4, 5, 6, 7, 8, and 9 significantly reduced the viral titer compared to control LNA gapmers. Of these, LNA gapmer 8 was the most effective in inhibiting JEV proliferation, followed by LNA gapmer 9. Interestingly, LNA gapmer 1 targeting two different regions was ineffective although it has more chances to bind JEV RNAs. LNA gapmers 2 and 3 targeting the 3′ UTR CS I region did not induce a significant reduction in viral titer compared to control LNA gapmers, which is inconsistent with the previous reports of the steric block ASOs using PMO and PNA chemistries [29,30,31]. 

The dose-dependent analysis further confirmed the antiviral effect of select LNA gapmers targeting the 3′ UTR stem-loop on suppressing JEV proliferation (Figure 2b). Vero cells were infected with the JEV JaGAr 01 strain at 0.05 MOI and then transfected with LNA gapmers 7, 8, and 9 at different concentrations of 0.05 to 5 µM. Those LNA gapmers showed a potent inhibitory effect on JEV proliferation in a dose-dependent manner. All LNA gapmers at 0.5 µM significantly reduced JEV titer (mean 0.7 × 10^6^ to 1.2 × 10^6^ PFU/mL) compared to a control LNA gapmer at 5 µM (mean 1.8 × 10^6^ PFU/mL). We also observed a significant increase in the antiviral activity for JEV proliferation when compared between 0.05 µM and 5 µM doses, or 0.5 µM and 5 µM doses for the three LNA gapmers (mean 2.7 × 10^5^ to 6.2 × 10^5^ PFU/mL at 5 µM). These results showed that the 3′ UTR stem-loop region is important in inducing the antiviral activity of LNA gapmers.

### 2.3. RNase H-Mediated Antiviral Mechanism of LNA Gapmers with Sequence and Modification Specificity

To examine the mechanism of viral RNA degradation through the reaction of RNase H to target RNA and LNA gapmer hybrids, we performed cell-based and biochemical RNA cleavage assays using mismatched (MM) LNA gapmers (Figure 3). The MM LNA gapmers were designed based on the original LNA gapmer 8; i.e., one or two nucleotides mismatched to the target sequence of JEV RNA were introduced in the LNA wings or RNase H-binding DNA region of LNA gapmer 8 (Figure 3a). We also prepared an ASO with no LNA modification, termed “all DNA”, to examine the importance of LNA modification in antiviral gapmers. In the cell-based mismatch test using Vero cells, we found the significantly impaired antiviral activity of MM LNA gapmers and the all DNA ASO, as represented by plaque assay (Figure 3b). Compared to the virus control, all MM LNA gapmers, except for MM2, and the all DNA ASO did not induce a significant reduction in JEV titers, unlike the original LNA gapmer 8. MM1 and 4, which have two mismatched nucleotides in the LNA wings or DNA region, showed a greater decline in antiviral activity than MM2 and 3 with a single mismatched nucleotide. We also observed a significant reduction in the ability of all MM LNA gapmers and the all DNA ASO to suppress JEV production compared to the original LNA gapmer 8.

In an RNA cleavage assay with the biochemical reaction of synthetic target JEV RNA to LNA gapmers and RNase H, we demonstrated that our LNA gapmers enable the degradation of the JEV RNA sequence in an RNase-H-dependent manner with time (Figure 3c, Appendix A). LNA gapmers 7, 8, and 9 induced a significant and quick degradation of the target JEV RNA sequence in 5 min after RNase H addition, as indicated by less than 50% reduction in the band intensity of the RNA/LNA gapmer complex. These RNA/LNA gapmer complex bands almost completely disappeared in 30 min after adding RNase H. It was also confirmed in an experiment in the presence or absence of RNase H tested for 120 min that RNase H is essential for degrading synthetic JEV RNA with LNA gapmers (Appendix A). 

The RNA cleavage assay using MM LNA gapmers and the all DNA ASO revealed that the activities of the LNA gapmer in binding to and degrading the synthetic JEV RNA are remarkably reduced due to the mismatched nucleotides and LNA removal (Figure 3d,e and Appendix A). MM1 and MM4 LNA gapmers, containing two mismatched nucleotides in the LNA wings and RNase H-binding DNA region, respectively, and the all DNA ASO did not form the complex with synthetic JEV RNA (Figure 3d and Appendix A). In contrast, although MM2 and MM3 LNA gapmers, with a single mismatched nucleotide in an LNA wing forming the complex with synthetic JEV RNA, they showed impaired activity of the RNA degradation (Figure 3e and Appendix A), i.e., a 50% disappearance of the MM2 and MM3 complex bands was observed in 10 and 30 min, respectively; these times were longer than the 5 min of the original LNA gapmer 8, as shown in Figure 3c. Importantly, the observation from the RNA cleavage assay was consistent with that from the cell-based assay (Figure 3b), indicating that JEV RNA-targeted LNA gapmers exert the antiviral activity with RNase H-mediated RNA degradation in a sequence- and modification-specific manner. 

### 2.4. Antiviral Efficacy of JEV RNA-Targeted LNA Gapmers in a Human Neuroblastoma Cell Line

As shown in Figure 4, we examined the antiviral effect of LNA gapmers on the proliferation of the JEV JaGAr 01 strain in human neuroblastoma SK-N-SH cells as an in vitro model for JEV infection, which mainly affects neurons in the central nerve system (CNS) tissue [23]. The LNA gapmers targeting JEV RNA effectively inhibited the proliferation of the JaGAr 01 strain in SK-N-SH cells compared to the control LNA gapmers (Figure 4a). LNA gapmers 7 and 8 showed significant inhibition of JEV proliferation at the lowest concentration of 0.05 µM (means 6.9 × 10^5^ and 5.5 × 10^5^ PFU/mL, respectively) compared to control LNA gapmers at the same concentration (mean 1.3 × 10^6^ PFU/mL). Overall, LNA gapmers 7, 8, and 9 exhibited antiviral efficacy for JaGAr 01-infected SK-N-SH cells in a dose-dependent manner. Increasing LNA gapmers from 0.05 µM to 0.5 µM significantly enhanced their inhibitory effect on JEV proliferation in human cells (from mean 5.5–8.7 × 10^5^ to 0.8–1.3 × 10^5^ PFU/mL, respectively). The dose-responsiveness of LNA gapmer 7 to the JEV inhibition was higher than that of the other LNA gapmers, as represented by a significant increase in the activity at 0.5 µM compared to 0.25 µM. Interestingly, human neuroblastoma cells exhibited a more sensitive reaction to the LNA gapmer treatment than Vero cells derived from a monkey kidney; in Vero cells tested with the identical method, a similar effect was observed at the ten times higher concentration of 5 µM (Figure 2b). 

Next, to investigate whether LNA gapmers can decrease the expression levels of JEV RNA in SK-N-SH cells, we performed RT-qPCR using a specific primer set to amplify the region that LNA gapmers are supposed to disrupt (Figure 4b,c). RNA extracts were prepared from JaGAr 01-infected SK-N-SH cells used in the plaque assay described in Figure 4a. As shown in Figure 4c, treatment with LNA gapmer 7, 8, or 9 significantly decreased the expression levels of JEV RNA by 82–94% and 88–95% at 0.25 µM and 0.5 µM, respectively, compared to control LNA gapmers at the same concentrations. LNA gapmer 8 exhibited the highest inhibitory effect on JEV RNA replication. While LNA gapmers 7 and 8 with greater efficiency showed no dose-dependent effect, effectiveness with an increase in concentration was found in LNA gapmer 9. It is worth noting that the reactivity of JEV to LNA gapmers at RNA levels observed here was consistent with the efficiency at suppressing viral production represented by the plaque assay (Figure 4a). These results support the antiviral mechanism of LNA gapmers via viral RNA degradation.

### 2.5. Cytotoxic Effect of LNA Gapmers 

To evaluate the cytotoxicity of LNA gapmers in human neuroblastoma SK-N-SH cells, we measured the cell viability after transfection with LNA gapmers (Figure 5). SK-N-SH cells were treated using LNA gapmers with an identical procedure to that conducted for Figure 4; the cells were incubated with a medium containing LNA gapmers at different concentrations of 0.05 to 1 µM and a vehicle of Lipofectamine RNAiMAX for four hours, and then with a maintenance medium without the agents until the measurement of cell viability using the Cell Counting Kit-8 (CCK-8; Dojindo Laboratories, Kumamoto, Japan). Compared to the vehicle control, three LNA gapmers at all concentrations tested, except for LNA gapmer 8 at one µM (approximately 70%), did not significantly reduce the cell viability. In the cell-based evaluation, as shown in Figure 4, SK-N-SH cells were treated with LNA gapmers at the range from 0.05 to 0.5 µM, at which apparent cytotoxicity was undetectable in the CCK-8 assay. The CCK-8 assay result observed here supports that the reduction in viral titer by LNA gapmer treatment is not due to cell decrease caused by the cytotoxicity of LNA gapmers.

### 2.6. In Silico Analysis of the Sequence-Dependent Off-Target Effects of LNA Gapmers 

Using GGGenome, we assessed the off-target binding possibility of JEV RNA-targeted LNA gapmer sequences to the human genome and RNA (Table 2). The analysis revealed no complementary regions of human DNA and RNA perfectly matched to our LNA gapmer sequences. One and four complementary sequences with one mismatch or gap to 17- and 16-mer LNA gapmers, respectively, were found in the human genome DNA; they were assigned to *ACBD6*, *HECW1*, and non-coding DNA regions. It is of note that no complementary sequences with one mismatch or gap to JEV RNA-targeted LNA gapmers were calculated in the human RNA, similar to control LNA gapmers that were purposefully designed not to be complementary to human RNA sequences. For the LNA gapmers composed of 17-mer (7 and 8) and 16-mer (9), we found a total of 10 and 81 complementary sequences with two mismatches or gaps in human RNAs, respectively, which involve mRNA, non-coding RNAs, and others (Appendix A). Overall, we confirmed that the longer LNA gapmers 7 and 8 have fewer chances of off-target binding to human RNAs compared to 16-mer LNA gapmer 9. 

### 2.7. Potential Applicability of Antiviral LNA Gapmers to JEV Genotypes I–V

As five JEV genotypes (G I to V), defined by their unique sequences, are prevalent worldwide [34], developing a drug applicable to various JEV strains is required. To examine if our LNA gapmers can apply to different genotypes, we collected a total of 292 JEV strain sequences, including 288 from the NIH/NIAID Virus Pathogen Resource (ViPR) [35] and 4 from wild-type strains determined in this study. Then, we comprehensively analyzed the conservation of their 3′ UTR stem-loop region targeted by the LNA gapmers (Figure 6). The analysis revealed that the LNA gapmer target region is highly conserved in JEV genotypes I, III, IV, and V, but not in genotype II. The JEV RNA sequence targeted by LNA gapmers 7 and 8 was conserved in more than 97% of the genotypes I, III, and IV. We also confirmed high conservation (>97%) of the LNA gapmer 9 target region in the genotypes of I, III, IV, and V. Secondary structure analysis using the RNAfold program predicted the conserved 3′ UTR stem-loop structure formation in five different genotypes, as shown by a representative strain of each genotype (Appendix A) [36]. The conserved primary and secondary structures observed here indicate the broad applicability of our LNA gapmers to different JEV strains.

### 2.8. Antiviral Efficacy of LNA Gapmers against Different JEV Strains of Genotypes I and III

JEV genotypes I and III are currently predominant in Asia [34]. To validate the potential applicability of our LNA gapmers to different JEV wild-type strains, as predicted above, we empirically tested the inhibitory effect on the proliferation of two genotype I strains (J-8-170-B and J-8-194-S) and two genotype III strains (Mie44-1 and AS6). We confirmed the strains’ genotypes by sequencing the envelope protein-encoding gene following the determination of sequence conservation of the LNA gapmer target region in the four wild-type strains (Figure 7a). Based on the target sequence conservation, we analyzed the antiviral efficacy of LNA gapmers 7 and 9, which had lower toxicity (Figure 5), against the four JEV wild-type strains in human neuroblastoma cells (Figure 7b–e). We observed an encouraging result, namely that compared to a control LNA gapmer (mean 9.4 × 10^5^–1.9 × 10^6^ PFU/mL), LNA gapmers 7 and 9 at 0.5 µM significantly inhibited the proliferation of all four wild-type strains (mean 4.5 × 10^4^–1.2 × 10^5^ PFU/mL), regardless of the genotype difference. 

## 3. Discussion

Japanese encephalitis is one of the most important causes of viral encephalitis, and is recognized as a serious public health issue for residents in Asia and travelers to the endemic areas [23]. Although vaccine preventable, the development of specific therapies is required for constant threats of fatal or severe cases and the increase in the infection risk of the human population [37]. In the context of the current lack of a cure for Japanese encephalitis [23,38], the present study is the first to demonstrate the antiviral efficacy of RNA-targeted LNA gapmers against JEV infection. Through in vitro screening of LNA gapmers designed on the plus strand RNA, we identified an effective region, a 3′ UTR stem-loop region of the JEV RNA genome, for LNA gapmers to inhibit JEV proliferation. This study also revealed an antiviral mechanism in which LNA gapmers induce the RNase H-dependent degradation of JEV RNA and antiviral activity in a sequence- and modification-specific manner. Given a proven mechanism, we demonstrated a beneficial effect of our LNA gapmers on suppressing JEV progeny production and JEV RNA replication in human neuroblastoma SK-N-SH cells as a JEV infection model. Comprehensive database analyses showed that our LNA gapmers have a low possibility of sequence-dependent off-target effects on human RNAs. In addition, the broad-spectrum antiviral potential was computationally and empirically supported with the highly conserved target JEV RNA sequence and suppression of the viral progeny production in four JEV wild-type strains, respectively. This work represents the first step toward developing antiviral LNA gapmer therapy for JEV and other flavivirus infections. 

In this study, in vitro screening revealed that all six LNA gapmers designed on the 3′ UTR stem-loop region effectively inhibit JEV proliferation (Figure 2). In contrast, LNA gapmers targeting CS I, an essential motif for the replication, were ineffective, unlike previous reports with steric block ASOs [29,30,31]. This antiviral effect of the JEV 3′ UTR stem-loop region targeted in this study is also supported by a previous study using steric block ASOs composed of peptide-conjugated PNAs [31]. Another study using steric block ASOs modified with peptide-conjugated PMOs also reported that the propagation of a flavivirus, dengue virus, can be suppressed by targeting the 3′ UTR stem-loop region, which is nearly identical to the JEV one tested in this study [39]. A recent study has reported that LNA gapmers targeting a 3′ UTR stem-loop motif effectively inhibit SARS-CoV-2 growth [8]. In addition, other studies indicate that LNA gapmers targeting stem-loops formed in the ORFs and 5′UTR induce antiviral activity against SARS-CoV-2 and influenza A virus [7,9,11]. As shown in our secondary structure prediction (Figure 1c and Appendix A), the JEV 3′ UTR stem-loop region formed a single-stranded structure to which LNA gapmers are more likely to bind. In contrast, the 3′ UTR CS I region appeared to be folded into intricate structures. This difference in predicted structures may partially explain why the stem-loop-targeted LNA gapmers are more effective than the CS I-targeted LNA gapmers. Therefore, the stem-loop structures of viral RNAs can become a basis for antiviral ASO design while considering ASO types (blocking or degrading RNAs), their chemistry backbones, and viral RNA nature, such as functions and sequence conservation. Optimization of the length and proportion of LNA gapmers can further enhance antiviral activity. Meanwhile, other regions targeted by LNA gapmers must be comprehensively explored to obtain more benefits [6,7,9].

As shown in Figure 3, we demonstrated an underlying mechanism of action of viral RNA-targeted LNA gapmers where RNase H-mediated degradation of JEV RNA induces the antiviral activity to suppress JEV propagation. The cell-based assay using LNA gapmers having mismatched nucleotides showed a significant reduction in antiviral activity (Figure 3a,b). LNA gapmers containing two mismatches in LNA wings or RNase H-binding DNA region and all DNA ASO almost completely abrogated the activity for reducing JEV titers. Moderate impairment of LNA gapmer activity resulted from a single mismatch in their sequences. It is noted that the present cell-based assay result was consistent with that of biochemical RNA cleavage assay, indicating different impairment mechanisms depending on the number of mismatched nucleotides assigned (Figure 3d,e), i.e., the inactivation of the antiviral effect of LNA gapmers containing two mismatches and the all DNA ASO is attributed to the complete loss of the ability to bind to the target JEV RNA sequence. In contrast, the decrease in the antiviral activity of LNA gapmers with a single mismatch comes from the reduced degradation rate of the target RNA. This action mechanism of the viral RNA-targeted LNA gapmer was also supported by the fact that the original LNA gapmers degrade the target JEV RNA sequence in the presence of RNase H with time (Figure 3c, Appendix A). To our knowledge, the present study using mismatched LNA gapmers is the first to provide evidence associating the antiviral activity of LNA gapmers with their ability to degrade viral RNA sequences. This antiviral mechanism via LNA gapmer-mediated viral RNA cleavage was further supported by the result showing that LNA gapmers inhibited JEV progeny production accompanied by the reduction in JEV genomic RNA levels in SK-N-SH cells (Figure 4). 

RNase H-mediated degradation of hepatitis C virus (HCV) subgenomic replicon RNA [40] and synthetic SARS-CoV-2 RNA [8] by LNA gapmers have previously been reported. Laxton et al. showed a similar result to ours; the ability of LNA gapmers to inhibit HCV replicon RNA production is reduced by one mismatched nucleotide and abolished by two to three substitutions [40]. However, the inhibitory effect of LNA gapmers on yielding infective viruses remains to be examined. Regarding this issue, the present study, using the combination of cell-based and RNA cleavage assays, found that LNA gapmers inhibit the production of infective JEV progenies in a sequence- and modification-dependent manner. Our results also suggest an essential aspect in LNA gapmer design, namely that the sequence specificity of LNA wings and the RNase H-binding DNA region is equally critical for antiviral activity. 

In the development of therapy for JEV infection, examining the antiviral activity of candidate drugs in human neuronal cells is essential because JEV mainly affects and spreads in brain tissue via producing progeny virions. The antiviral potential of JEV RNA-targeted ASOs has been tested in various animal cells but has not yet been examined in human cells [29,30,31]. In this study, we were successful in providing a proof-of-concept demonstration that LNA gapmers effectively inhibit JEV proliferation in a human neuroblastoma SK-N-SH cell line as an infection model of JEV (Figure 4). Similar to the test in Vero cells, LNA gapmers exhibited antiviral efficacy in human SK-N-SH cells in a dose-dependent manner. A notable difference was that a significant efficacy in SK-N-SH cells was induced at 0.05 μM, ten times lower than the 0.5 μM found in Vero cells. Although the effect of use in a lipofection agent for LNA gapmer transfection is considered, one possibility may be a difference in the cellular uptake efficiency of LNA gapmers between SK-N-SH and Vero cells. Developing efficient drug delivery systems to target tissues, particularly nerve systems [41], is one of the major issues facing the field of ASO therapies [42]. Further investigations are required to understand and improve the delivery efficiency of LNA gapmers into viral-infected neuronal cells. 

Regarding safety, we encouragingly observed that the inhibition of JEV propagation in SK-N-SH cells was achieved using our LNA gapmers at the concentrations showing no apparent cytotoxicity as detected by the cell viability (Figure 5). Although one of our LNA gapmers showed a significant reduction in cell viability of SK-N-SH cells treated at the highest 1 μM, this concentration was four times higher than 0.25 μM at which inhibition efficiency for JEV-infected SK-N-SH cells almost reaches a plateau (Figure 4). Other JEV-RNA-targeted and control LNA gapmers were not toxic in SK-N-SH cells at any concentration tested. The differences between LNA gapmer 8 showing cytotoxicity and nontoxic LNA gapmer 7 are their proportion and position of LNA; while LNA gapmer 8 is composed of the same 17-mer as LNA gapmer 7, it has one more LNA in the 5′ LNA wing and the triplet LNA is moved inward by one nucleotide in the 3′ LNA wing (Table 1). In vivo studies report adverse effects similar to our result that altering LNA proportions in LNA gapmers with the same length results in different degrees of hepatotoxicity in mice, as detected by serum chemistry [43,44]. Adverse effects of ASOs with LNA and phosphorothioate modifications involve hepatotoxicity, renal toxicity, neurotoxicity, and coagulation inhibition [45]. These toxicities of ASOs, including LNA gapmers, are primarily classified into ASO hybridization/sequence-dependent or independent ones [45]. As shown in Table 2, the present database analysis predicted no human RNAs perfectly matched or one-mismatched complementary to JEV RNA-targeted LNA gapmer sequences, suggesting that sequence-dependent off-target effects are unlikely to occur in our antiviral LNA gapmers. In contrast, hybridization/sequence-independent toxicity results from the ASO chemical backbone and its accumulation in high-exposure organs [45], for which the effects need to be examined in animal models. Several design algorithms and safety assessments of LNA gapmers have been proposed to prevent potential toxicity [45,46,47,48,49]. Although becoming more complex considering viral pathology, those preventive approaches developed for LNA gapmers targeting cell-derived RNAs can be applied to designing those targeting viral RNAs, as we showed here using GGGenome.

Unlike human genetic diseases caused by a single causative mRNA, viral RNA-targeted LNA gapmers need to possess broad-spectrum activity across JEV genotypes and strains classified as different RNA genome sequences. Although we initially designed LNA gapmers referring to the 3′ UTR sequences of the JaGAr 01 strain and some vaccine strains, a comprehensive analysis using the virus genome database and JEV RNA sequences determined in this study revealed that our lead LNA gapmers targeting the stem-loop region are amenable to most registered JEV strains (96% out of the 292 strains) in theory (Figure 6). Notably, all five genotypes highly conserved the stem-loop conformation formed with the LNA gapmer target sequences (Appendix A). The broad-spectrum potential of viral RNA-targeted therapies to different JEV genotypes/strains was previously reported in small interfering RNAs [50] and lentiviral small hairpin RNAs [51,52]. However, this possibility remained unclear in JEV RNA-targeted ASOs. Here, we first showed that a single ASO targeting a conserved region of JEV genomic RNA can be applied to different JEV strains of the predominant genotypes I and III in Asia (Figure 7). Meanwhile, the number of cases of genotypes II, IV, and V has been increasing in recent years [34,53,54]. Although empirical examinations against such rare genotypes are needed, it is expected that our LNA gapmers have the potential to become a therapeutic strategy covering a variety of JEV genotypes and strains based on the target sequence conservation. Even though in some JEV strains the target 3′ UTR stem-loop region is not perfectly complementary to our LNA gapmers, chemically synthesized LNA gapmer sequences can be flexibly adjusted to different JEV RNA sequences of such exceptional strains. Given this flexibility, it is emphasized that our LNA gapmers could exert antiviral activity against other flavivirus species, such as YFV, WNV, and DNV, because the 3′ UTR stem-loop region tested in this study is highly conserved among those flaviviruses [31,55].

Regarding the clinical application of LNA gapmer therapy for JEV infection, it is necessary to examine in vivo efficacy and safety of the drug, particularly a drug delivery system to the CNS, which is critically affected by JEV. A major challenge in developing ASO therapeutics, including LNA gapmers, is to find an efficient way to deliver agents to the target brain crossing the blood–brain barrier (BBB). Although LNA gapmers can be distributed to various tissues, an issue is that they have a limited ability to reach the brain [56]. An intrathecal injection may be an option to overcome this biological barrier, as an ASO drug called nusinersen, for the neuromuscular disease spinal muscular atrophy, is in clinical application via this local administration [57]. As a more promising solution, a recent study has reported an advanced technology that gives LNA gapmers a lipid–ligand conjugate and a DNA/RNA heteroduplex oligonucleotide (HDO) form, enabling them to cross the BBB and efficiently reach neurons of the brain [58]. This CNS approach may be more reasonable for clinical use in individuals with Japanese encephalitis. In terms of the infection process, because the BBB is disrupted and its permeability increases with the progression of JEV infection [59], intravenous administration of LNA gapmers might allow JEV-infected brain tissue of patients to be reached and treated at a particular clinical stage. Together with the LNA gapmer optimization in human cells, these clinical potentials must be further investigated using appropriate animal models and JEV strains [33].

In conclusion, the present study demonstrated the therapeutic potential of LNA gapmers for efficiently inhibiting JEV progeny production through RNase H-mediated viral RNA degradation in a sequence- and modification-dependent manner in vitro (Figure 1a). Consequently, we have identified that LNA gapmers may become a promising antiviral candidate for JEV. We also revealed that the conserved JEV 3′ UTR stem-loop region is a compelling target for developing antiviral LNA gapmers. Furthermore, we confirmed the broad-spectrum potential of our LNA gapmers to various JEV strains. These results emphasize that targeting JEV RNA using LNA gapmers can become a therapeutic approach for treating JEV infection. Future works to increase the antiviral potency and applicability will involve the optimization of LNA gapmers and understanding molecular mechanisms in degrading viral RNAs, such as genomic RNAs and minus-strand RNAs. Our findings support the further development of LNA gapmer therapeutics against pathogenic flaviviruses including JEV. 

## 4. Materials and Methods

### 4.1. Cells and Viruses

Vero cells and SK-N-SH cells were obtained from the RIKEN BioResource Research Center (BRC). Vero cells were cultured with growth medium (GM): Eagle’s minimal essential medium (EMEM; Wako, Osaka, Japan) containing 10% heat-inactivated fetal bovine serum (FBS; GE health care, Chicago, IL, USA) and 0.5% antibiotics (50 units/mL penicillin and 50 µg/mL streptomycin [Wako]) at 37 °C in 5% CO_2_. SK-N-SH cells were cultured with antibiotics-free Minimum Essential Medium α (MEMα; Wako) containing 10% FBS at 37 °C in 5% CO_2_. Five JEV strains used in this study were passaged three times in Vero cells: JaGAr 01 (genotype III, accession no. AF069076.1), J-8-170-B (genotype I, LC777833), J-8-194-S (genotype I, LC777834), Mie44-1 (genotype III, LC777831), and AS6 (genotype III, LC777832). The supernatant containing viruses was aliquoted and stored at −80 °C until use. The titers of virus stocks were quantified by plaque assay to determine a multiplicity of infection (MOI). 

### 4.2. JEV RNA Secondary Structure Prediction

Complete genome sequences of JEV strains belonging to either of five genotypes were obtained from National Center for Biotechnology Information (NCBI) GenBank: SH 53 (GI, JN381850.1), FU (GII, AF217620.1), JaGAr 01 (GIII, AF069076.1), GP78 (GIII, AF075723.1), JKT6468 (GIV, AY184212.1) and Muar strain (GV, HM596272.1). JEV RNA secondary structures of 3′ UTR were predicted using the RNAfold program (http://rna.tbi.univie.ac.at/cgi-bin/RNAWebSuite/RNAfold.cgi accessed on 29 November 2022) [36].

### 4.3. LNA Gapmer Design and Synthesis

LNA gapmer 1 targeting two different regions with the same sequence in the 3′ UTR of the JaGAr 01 strain was designed and synthesized by QIAGEN (ID: LG00204874) for a preliminary test. LNA gapmers 2 to 9 were designed based on the sequences and predicted structures of the 3′ UTR of the JaGAr 01 strain, referring to those of other strains (Figure 1c and Table 1). Control LNA gapmers were designed not to be complementary to JEV RNA and the human genome, considering LNA/DNA proportion. Of these, control LNA gapmer 4 was from a previous study [60]. Mismatched LNA gapmers were designed based on the most effective LNA gapmer 8. The substituted bases and their positions in mismatched LNA gapmers were determined considering the base transversion and functional regions. LNA gapmers and the all DNA ASO involving phosphorothioate modification were synthesized with HPLC grade by Ajinomoto Bio-Pharma Services GeneDesign, Inc. (Osaka, Japan).

### 4.4. Transfection

Vero cells were seeded at a density of 1.0 × 10^5^ cells/well in a 24-well plate and incubated at 37 °C in 5% CO_2_ for 24 h to form approximately 95% confluent cell monolayer. SK-N-SH cells seeded at a density of 1.4 × 10^5^ cells/well in a 24-well plate were grown for 72 h to more than 95% confluent. Cells were infected with JEV at the MOI of 0.1 and 0.05 for in vitro screening in Vero cells and other experiments, respectively, for 1 h with tilting every 15 min. The virus solution was diluted with maintenance medium: EMEM and MEMα containing 2% FBS for Vero and SK-N-SH cells, respectively. LNA gapmer transfection was performed with lipofection with Lipofectamine RNAiMAX (Thermo Fisher Scientific, MA, USA). In brief, during the inoculation of the virus, an 80 µL mixture of LNA gapmer from 0.15 to 15 µM and 3 µL Lipofectamine RNAiMAX was prepared in Opti-MEM (Thermo Fisher Scientific). To obtain final concentrations from 0.05 to 5 µM of LNA gapmers to be tested, the LNA mixture was diluted three times with EMEM and MEMα with 5% FBS for Vero and SK-N-SH cells, respectively. After removing the virus solution, cells were transfected with 230 µL of the LNA mixture prepared above for 4 h. Following the transfection, cells were washed once with a medium with 5% FBS and then maintained in a maintenance medium with 2% FBS. The supernatants were collected at 29 and 24 h post-infection (hpi) of JaGAr 01 for plaque assay in a screening test with Vero cells and other experiments, respectively. The supernatants of SK-N-SH cells infected with four other JEV strains were obtained at 48 hpi. Virus titers were expressed as the number of plaque-forming units (PFU) per milliliter. A schematic of the transfection is shown in Appendix A.

### 4.5. Plaque Assay

Vero cells were seeded on 12-well plates at 2.0 × 10^5^ cells/well density in GM: EMEM supplemented with 10% FBS and 0.5% antibiotics. Following the removal of GM, over 95% confluent Vero cells were infected with 10-times serial dilutions of viral culture supernatants, which were prepared in the maintenance medium with 2% FBS and 0.5% antibiotics, for 1 h at 37 °C in 5% CO_2_ tilting every 15 min. Following the infection, cells were washed once with the maintenance medium and then overlaid with 1 mL of a semisolid medium (1% methylcellulose (Wako), 2% FBS, and 0.5% penicillin/streptomycin in EMEM). After incubation for 72 h, 500 µL of 10% formalin neutral buffer solution (Wako) was added to each well to fix cells, and 12-well plates were placed under UV light for 30 min. After the cell fixation, cells were washed with distilled water and stained with 0.5% crystal violet (Merck, Darmstadt, Germany) for the counting of plaque numbers and calculating virus titers as PFU/mL. 

### 4.6. RNA Cleavage Assay

A target 17-mer JEV RNA sequence complementary to 3′ UTR stem-loop-targeted LNA gapmers was synthesized with HPLC grade by Ajinomoto Bio-Pharma Services GeneDesign, Inc. LNA gapmers (1.5 µM), synthesized JEV RNA (6 µM), and *E. coli* RNase H (50 units/mL; New England Biolabs, Ipswich, MA, USA) were mixed and incubated in 1 × RNase H reaction buffer at 37 °C for 120 min. The 20 µL reaction mixture was collected at 0, 5, 10, 30, 60, and 120 min after the reaction. The collected reaction mixture was immediately quenched by adding 1 µL of 0.5 M EDTA (NIPPON GENE, Tokyo, Japan) at pH 8.0 and denatured with an equal amount of 21 µL formamide (Wako). The samples were immediately used for a subsequent procedure or stored at −80 °C until use. They were mixed with an equal volume of Novex^TM^ TBE-Urea sample buffer (Thermo Fisher Scientific), heated at 70 °C for 4 min, and immediately placed on ice. Aliquots having a volume of 20 µL and 12 µL microRNA marker (New England Biolabs) were submitted to a Mini-PROTEAN 15% TBE-urea gel with ten wells (Bio-Rad, Santa Rosa, CA, USA) and electrophoresed at 200 V for 30 min. The gel was stained by SYBR® Green II (Thermo Fisher Scientific), and images were captured using the Gel Doc^TM^ EZ system (Bio-Rad). The band intensities were quantified using ImageJ software (version 1.53k; NIH).

### 4.7. Sequencing and Quantitative Real-Time RT-PCR

JEV RNA sequences targeted by LNA gapmers or encoding envelope protein [61] were amplified using PrimeScript One Step RT-PCR Kit Ver.2 (Takara, Kusatsu, Japan) according to the manufacturer’s instruction and determined in Azenta Life Sciences (Tokyo, Japan). Primers used are listed in Table 3.

The relative expression levels of JEV RNA were analyzed using the comparative Ct method. The primer set to amplify LNA gapmer target region on JaGAr 01 strain genomic RNA was designed using Primer 3 Plus. As a reference gene, human *GAPDH* mRNA levels were measured using a primer set as previously reported [62]. To use the primer set for the comparative Ct method, we confirmed the absolute slope value of <0.1, plotting the log RNA dilution and the delta Ct values of each primer set [63]. Cells were harvested at 24 hpi using 1 mL of TRIzol^TM^ Reagent (Thermo Fisher Scientific) per well and stored at −80 °C until use. Then, RNA was extracted using the RNeasy Mini Kit (QIAGEN). RNA concentrations were quantified using Nano drop (Thermo Fisher Scientific) and normalized to 20 ng/µL. Real-time RT-PCR was performed with the Thermal Cycler Dice^®^ Real Time System III (Takara) using One Step TB Green^®^ PrimeScript^TM^ PLUS RT-PCR Kit (Takara). The total reaction volume was 25 µL with 12.5 µL of 2 × One Step TB Green RT-PCR Buffer, 0.5 µL of PrimeScript PLUS RTase Mix, 0.4 µM each of the forward and reverse primers, 1.5 µL of TaKaRa Ex Taq HS Mix, and 2 µL of the RNA. The amplification condition was 5 min at 42 °C and then 10 s at 95 °C, followed by 40 cycles of 5 s at 95 °C and 30 s at 60 °C. The expression levels of JEV RNA were normalized to human *GAPDH* mRNA levels. 

### 4.8. Cytotoxicity Assay 

SK-N-SH cells were plated at 0.4 × 10^5^ cells/well on 96-well plates in GM: MEMα containing 10% FBS and cultured for 72 h. For lipofection, a 240 µL mixture of LNA gapmers at the final concentration from 0.05 to 1 µM and 3 µL of Lipofectamine RNAiMAX was prepared with an identical procedure to that described above. Cells were transfected with 40 µL of the transfection mixture for 4 h. After transfection, cells were washed with MEMα containing 5% FBS once and incubated in 100 µL MEMα with 2% FBS for 18 h. After incubation, the CCK-8 assay was conducted according to the manufacturer’s instruction. A quantity of 10 µL of CCK-8 solution was added to each well; 96-well plates were incubated for 2 h. Then, the absorbance was measured using a microplate reader, Multiskan GO (Thermo Fisher Scientific), at 450 nm. The cell viability was calculated and represented by the percentage of the negative control treated with neither LNA gapmers nor Lipofectamine RNAiMAX as 100%.

### 4.9. Analysis for Potential Off-Target Binding of JEV RNA-Targeted LNA Gapmers to Human DNA and RNA

The number of complementary regions to LNA gapmer 7, 8, and 9 and control LNA gapmer sequences with or without mismatches or gaps in the human genome (Human genome, GRCh38/hg38 (Dec 2013)) and human RNA (RefSeq human RNA release 210 (Jan 2022)) was analyzed using GGGenome (https://gggenome.dbcls.jp/ja/ accessed on 29 November 2022). 

### 4.10. Conservation Analysis of LNA Gapmer Target Region in JEV Genotypes

A total of 292 JEV RNA sequences that LNA gapmers 7, 8, and 9 target were collected using ViPR (http://www.viprbrc.org/ accessed on 2 May 2022) and from the four JEV strains determined in this study (J-8-170-B, J-8-194-S, Mie44-1, and AS6). They were composed of 136, 3, 146, 4, and 3 strains of the five genotypes I to V, respectively. The conservation of the JEV RNA sequences complementary to the LNA gapmers was analyzed using Clustal Omega (https://www.ebi.ac.uk/Tools/msa/clustalo/ accessed on 2 May 2022).

### 4.11. Statistical Analysis

Data are expressed as the mean and SD or SE from at least three independent experiments. For statistical analyses, the PFU/mL was converted to logarithms and analyzed using one-way ANOVA followed by a post hoc Tukey–Kramer multiple comparison test using GraphPad Prism 8 (GraphPad Software, La Jolla, CA, USA). In RNA cleavage assay and cytotoxic assay, the Shirley−Williams test was performed using Statcel-the Useful Addin Forms on Excel-4th ed (OMS Ltd., Saitama, Japan). In quantitative real-time RT-PCR, delta Ct values were compared using Student’s t-test and one-way ANOVA followed by a post hoc Tukey–Kramer multiple comparison test in GraphPad Prism 8. The *p*-value < 0.05 was considered statistically significant. 

## Figures and Tables

**Figure 1 ijms-24-14846-f001:**
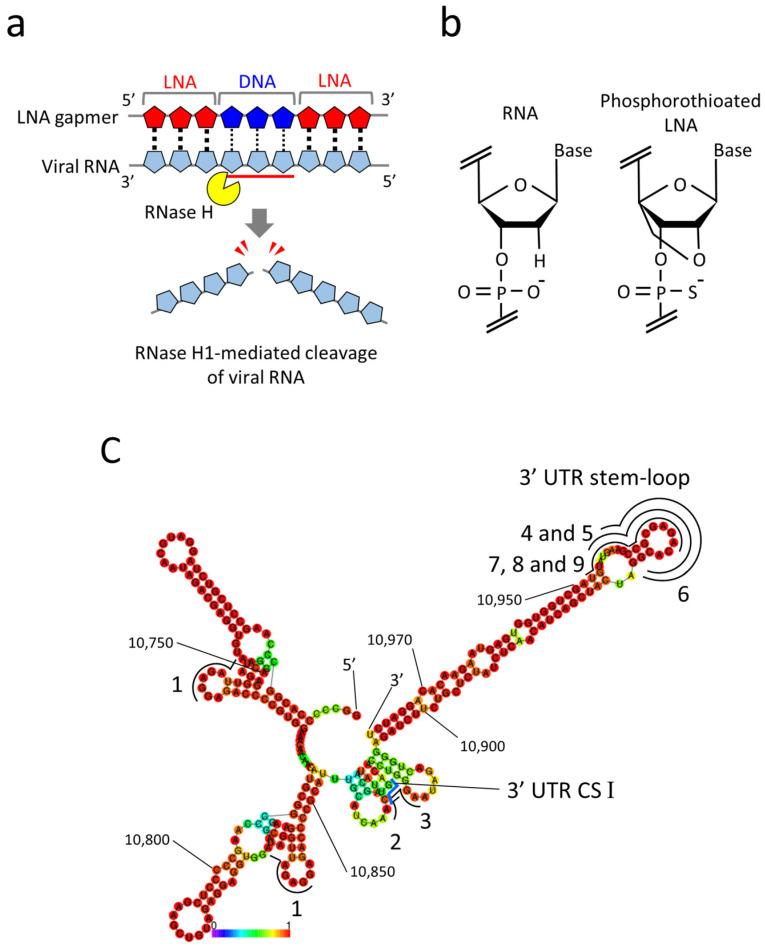
LNA gapmers targeting RNA secondary structures in the 3′ UTR of the JEV genomic RNA. (**a**) A mechanism of antiviral LNA gapmers. LNA gapmers bind to viral RNA by sequence complementarity and induce RNase H-mediated cleavage of viral RNA hybridized to the LNA gapmer DNA region. (**b**) Structures of RNA and LNA. LNA is characterized by an O2′-C4′-methylene linkage and phosphorothioate modification. (**c**) The RNA secondary structure of the partial 3′ UTR (10,700–10,977 nt) of the JEV JaGAr 01 strain (Accession no. AF069076.1) was predicted by the RNAfold program. The 3′ UTR conserved sequence (CS) I region is indicated with a blue line. The regions targeted by LNA gapmers are shown with black lines with numbers. Numbers in the smaller font refer to nucleotide position in the genome. A heat map color below the structure represents the base-pairing probabilities. The redder they are, the more likely they are to form the indicated secondary structures.

**Figure 2 ijms-24-14846-f002:**
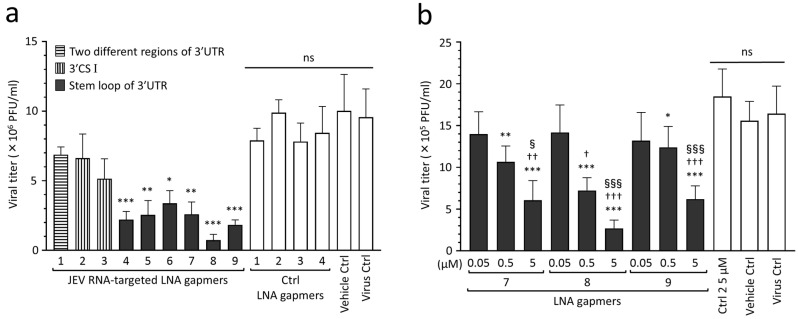
Inhibitory effect of JEV RNA-targeted LNA gapmers on JEV proliferation in Vero cells. (**a**) In vitro screening of designed LNA gapmers targeting the 3′ UTR of JEV RNA, as represented by plaque assay (PFU/mL). Vero cells were infected with JEV JaGAr 01 at 0.1 MOI and transfected with LNA gapmers at the concentration of 5 µM using a vehicle, Lipofectamine RNAiMAX (see Appendix A for the scheme). Data represent mean and standard deviation (SD) from three independent experiments. The statistical analysis using logarithmic values was performed by one-way ANOVA followed by Tukey–Kramer multiple comparison test: * *p* < 0.05, ** *p* < 0.01, *** *p* < 0.001 vs. four control (Ctrl) LNA gapmers; ns, no significant difference. (**b**) Dose-dependent effect of the LNA gapmers targeting the stem-loop region as measured by plaque assay. Vero cells infected with JEV at 0.05 MOI were transfected with LNA gapmers 7, 8, and 9 at the indicated concentrations. Data represent mean and SD from six independent experiments. * *p* < 0.05, ** *p* < 0.01, *** *p* < 0.001 vs. ctrl LNA gapmer 2 at 5 µM. ^†^ *p* < 0.05, ^††^ *p* < 0.01, ^†††^ *p* < 0.001 vs. 0.05 µM in the same LNA gapmer. ^§^ *p* < 0.05, ^§§§^ *p* < 0.001 vs. 0.5 µM in the same LNA gapmer.

**Figure 3 ijms-24-14846-f003:**
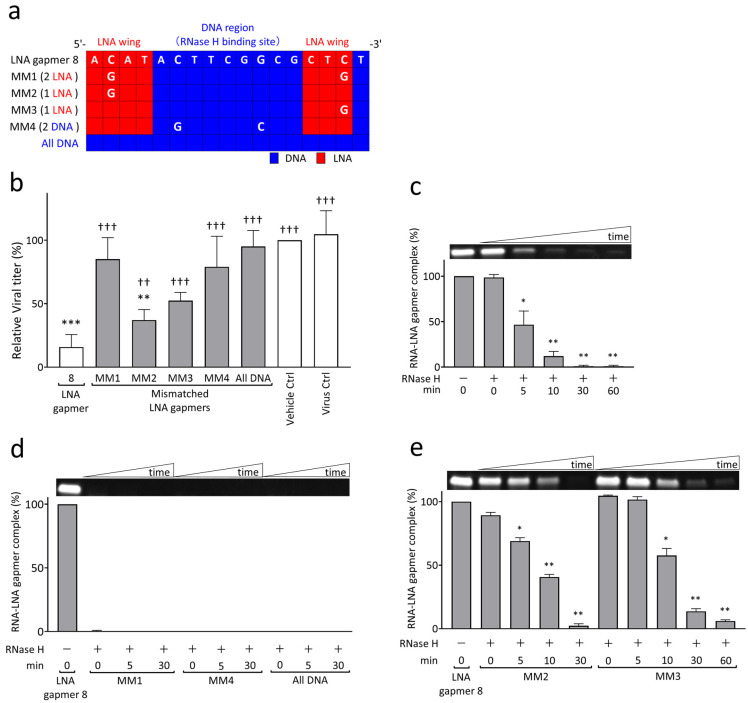
RNase H-mediated antiviral mechanism of JEV RNA-targeted LNA gapmers measured by cell-based and biochemical RNA cleavage assays using mismatched (MM) LNA gapmers. (**a**) MM LNA gapmer sequences used for the experiment. The mismatched nucleotides are shown as white letters compared to the original LNA gapmer 8. The red and blue squares indicate LNA and DNA, respectively. MM1, MM2, and MM3 contain two or one mismatched nucleotides in LNA wings. MM4 has two mismatched nucleotides in the RNase H-binding DNA region. All DNA has no LNA. (**b**) Attenuated inhibitory effect of MM LNA gapmers on JEV proliferation in Vero cells as represented by plaque assay. Vero cells infected with JEV JaGAr 01 at 0.05 MOI were treated with LNA gapmers at the concentration of 5 µM, and the viral titers (PFU/mL) were measured by plaque assay. Data represent mean and SD from four independent experiments. The result is shown as the percentage; vehicle control was defined as 100%. The statistical analysis using logarithmic values was performed by one-way ANOVA followed by Tukey–Kramer multiple comparison test: ** *p* < 0.01, *** *p* < 0.001 vs. virus control. ^††^ *p* < 0.01, ^†††^ *p* < 0.001 vs. LNA gapmer 8. (**c**) RNase H-mediated degradation of synthetic JEV RNA bound to LNA gapmer 8 over time as represented by RNA cleavage assay. A representative band image of the synthetic JEV RNA and LNA gapmer complexes is shown above the graph (an uncropped image is available in Appendix A). The negative control of RNase H (-) is considered as 100%. The graphs calculated with band intensities represent the mean and SD of three independent experiments. The results with LNA gapmers 7 and 9 and the uncropped images are available in Appendix A. (**d**,**e**) The impairment of sequence-specific degradation activity to JEV RNA by a few mismatched nucleotides in an LNA gapmer as represented by RNA cleavage assay: (**d**) the loss of binding ability to the target RNA by double mismatches or no LNA in an ASO and (**e**) the decreased ability to degrade the JEV RNA sequence by a single mismatch. The graphs with band intensities are represented as the percentage of 0 min in the absence of RNase H, defined as 100% (mean, SD of *n* = 3 in each). Shirley–Williams test was performed for statistical analysis: * *p* < 0.05, ** *p* < 0.01 vs. 0 min just after the RNase H addition.

**Figure 4 ijms-24-14846-f004:**
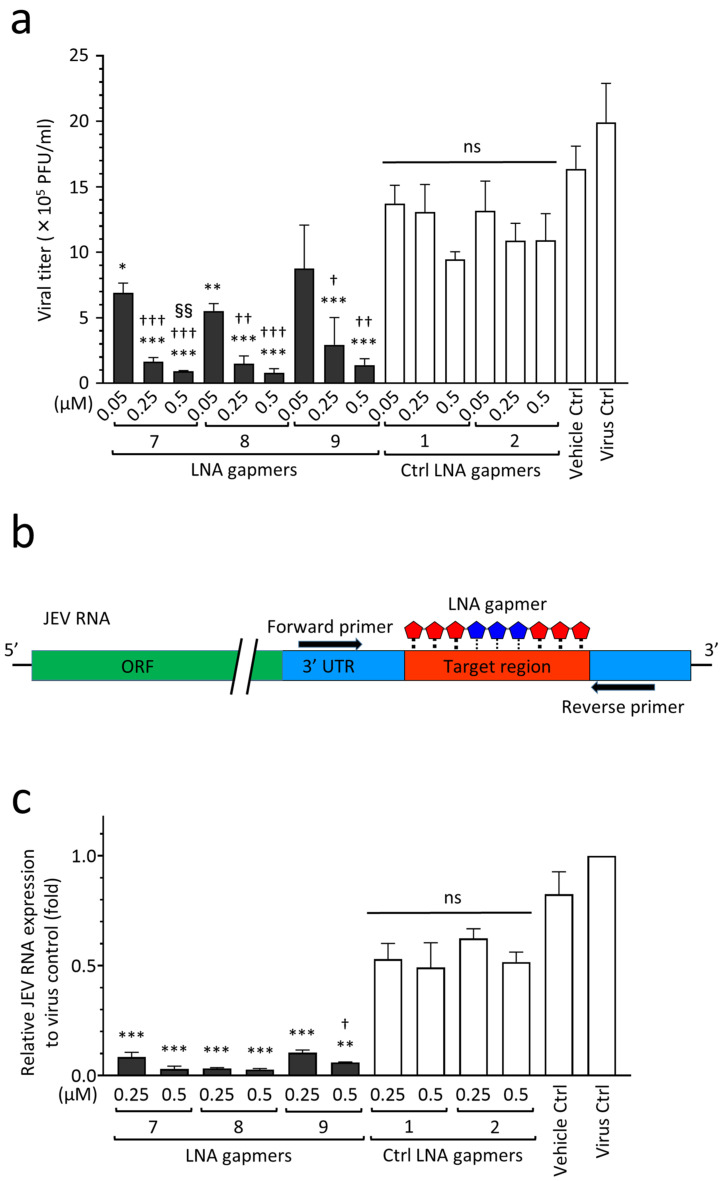
Antiviral activity of LNA gapmers against JEV progeny production and viral RNA replication in human neuroblastoma cells (SK-N-SH cells). (**a**) Inhibitory effect of LNA gapmers on the viral proliferation as measured by plaque assay. Following JEV JaGAr 01 infection at 0.05 MOI for 1 h, SK-N-SH cells were treated with JEV RNA-targeted (7, 8, and 9) or control (Ctrl 1 and 2) LNA gapmers in lipofection. Data represent mean and SD with PFU/mL from at least three independent experiments. Statistical analysis using logarithmic values was conducted using one-way ANOVA followed by Tukey–Kramer multiple comparison test: * *p* < 0.05, ** *p* < 0.01, *** *p* < 0.001 vs. control LNA gapmers at the same concentration as one in each LNA gapmer. ^†^ *p* < 0.05, ^††^ *p* < 0.01, ^†††^ *p* < 0.001 vs. LNA gapmer at 0.05 µM. ^§§^ *p* < 0.01 vs. LNA gapmer at 0.25 µM. ns, no significant difference. (**b**) Schematic representation of RT-qPCR using a primer set to examine the expression levels of LNA gapmer target JEV RNA region. (**c**) The relative expression levels of JEV RNA in SK-N-SH cells as measured by RT-qPCR with the comparative Ct method. The cell samples were the same as those used for plaque assay (**a**). Data represent mean and standard error (SE) relative to virus control (*n* = 3). Student’s t test and one-way ANOVA followed by Tukey–Kramer multiple comparison test were performed to compare between the different concentrations and in the same concentration, respectively. ** *p* < 0.01, *** *p* < 0.001 vs. control LNA gapmers at the same concentration as one in each LNA gapmer. ^†^ *p* < 0.05 vs. 0.25 µM of the same LNA gapmer.

**Figure 5 ijms-24-14846-f005:**
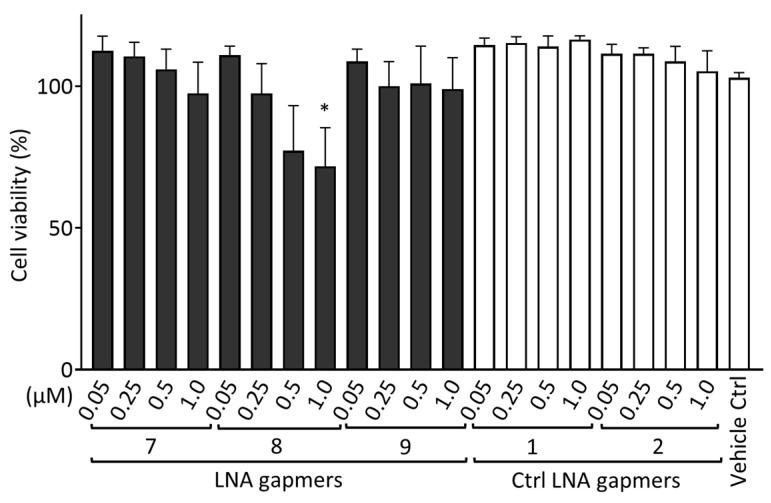
Cytotoxicity evaluation of LNA gapmers in human neuroblastoma SK-N-SH cells. Cells were transfected with LNA gapmers at different concentrations from 0.05 to 1 μM. After the treatment identical to the procedure of Figure 4 without JEV infection, the cell viability was measured using the CCK-8 assay. The cell viability of the negative control without any treatment was defined as 100%. Data represent the mean and SD of four independent experiments. Shirley−Williams test was performed for statistical analysis. * *p* < 0.05 vs. vehicle control (Ctrl).

**Figure 6 ijms-24-14846-f006:**
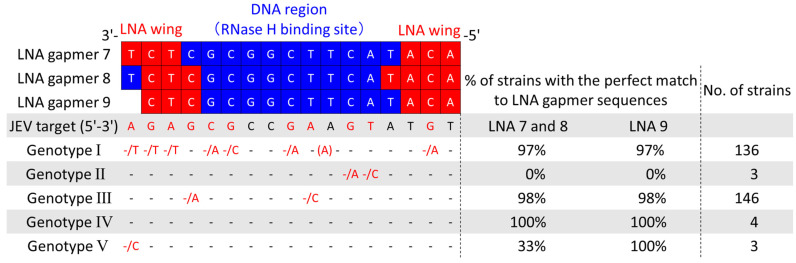
Comprehensive analysis of sequence conservation of JEV 3′UTR stem-loop targeted by the LNA gapmers using the virus pathogen resource (ViPR) and JEV RNA sequences determined in this study. Genomic RNA sequence data of a total of 292 JEV strains, which consist of five genotypes, were collected, and the conservation of the LNA gapmer target sequences was analyzed. JEV target indicates the RNA genome sequence from 10,932 to 10,948 nt of the JEV JaGAr 01 strain, targeted by the LNA gapmers tested in this study. Dashes indicate the same nucleotide as those in the JEV target. Nucleotides next to a slash indicate substituted nucleotides found in some JEV strains. A nucleotide in parenthesis is an inserted nucleotide. Red dashes and nucleotides indicate positions with substituted nucleotides.

**Figure 7 ijms-24-14846-f007:**
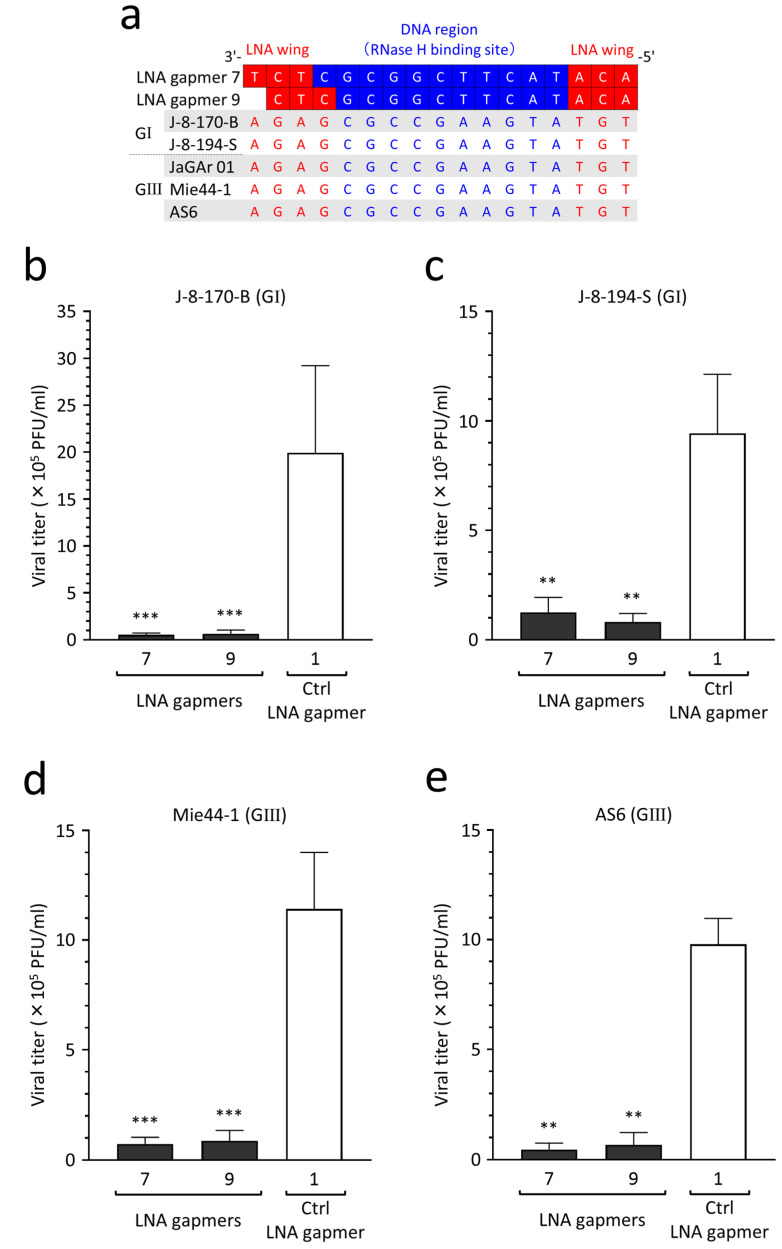
Inhibitory effect of LNA gapmers on the proliferation of JEV genotypes (G) I and III strains in SK-N-SH cells. (**a**) Complete conservation of LNA gapmer target region of the JEV strains used in this study. Red and blue letters indicate nucleotides complementary to the LNA wings and DNA region, respectively. (**b**–**e**) Antiviral efficacy of LNA gapmers at 0.5 µM against four JEV wild-type strains (0.05 MOI) in SK-N-SH cells. Viral titers (PFU/mL) were measured via plaque assay in Vero cells as described above: (**b**) J-8-170-B (GI), (**c**) J-8-194-S (GI), (**d**) Mie44-1 (GIII), (**e**) AS6 (GIII). Data represent mean and SD from three independent experiments. Statistical analysis using logarithmic values was conducted using one-way ANOVA followed by Tukey–Kramer multiple comparison test: ** *p* < 0.01, *** *p* < 0.001 vs. control LNA gapmer 1.

**Table 1 ijms-24-14846-t001:** LNA gapmer sequences and modifications used in this study.

Name	Sequence (5′→3′)	Target Regions of the JEV Genome✵(JaGAr 01, Accession No. AF069076.1)	Length (mer)
LNA gapmer 1	^m^C ^ T ^ ^m^C ^ t ^ a ^ a ^ c ^ c ^ t ^ c ^ t ^ a ^ G ^ T ^ ^m^C	10,749–10,763; 10,827–10,841	15
LNA gapmer 2	G ^ G ^ T ^ g ^ t ^ c ^ a ^ a ^ t ^ a ^ t ^ g ^ c ^ T ^ G ^ T	10,863–10,878	16
LNA gapmer 3	T ^ ^m^C ^ ^m^C ^ c ^ a ^ g ^ g ^ t ^ g ^ t ^ c ^ a ^ a ^ T ^ A ^ T	10,868–10,883	16
LNA gapmer 4	A ^ ^m^C ^ T ^ t ^ c ^ g ^ g ^ c ^ g ^ c ^ t ^ c ^ t ^ G ^ T ^ G	10,956–10,971	16
LNA gapmer 5	A ^ ^m^C ^ T ^ t ^ c ^ g ^ g ^ c ^ g ^ c ^ t ^ c ^ t ^ G ^ T ^ g	10,956–10,971	16
LNA gapmer 6	T ^ T ^ ^m^C ^ g ^ g ^ c ^ g ^ c ^ t ^ c ^ t ^ g ^ t ^ G ^ ^m^C ^ ^m^C	10,958–10,973	16
LNA gapmer 7	A ^ ^m^C ^ A ^ t ^ a ^ c ^ t ^ t ^ c ^ g ^ g ^ c ^ g ^ c ^ T ^ ^m^C ^ T	10,952–10,968	17
LNA gapmer 8	A ^ ^m^C ^ A ^ T ^ a ^ c ^ t ^ t ^ c ^ g ^ g ^ c ^ g ^ ^m^C ^ T ^ ^m^C ^ t	10,952–10,968	17
LNA gapmer 9	A ^ ^m^C ^ A ^ t ^ a ^ c ^ t ^ t ^ c ^ g ^ g ^ c ^ g ^ ^m^C ^ T ^ ^m^C	10,952–10,967	16
Control LNA gapmer 1	A ^ ^m^C ^ T ^ c ^ t ^ c ^ g ^ t ^ c ^ a ^ a ^ c ^ c ^ A ^ A ^ T	NA	16
Control LNA gapmer 2	G ^ T ^ A ^ a ^ c ^ t ^ c ^ g ^ t ^ c ^ g ^ t ^ a ^ A ^ ^m^C ^ A	NA	16
Control LNA gapmer 3	^m^C ^ G ^ A ^ a ^ t ^ a ^ g ^ t ^ t ^ a ^ g ^ t ^ a ^ G ^ ^m^C ^ G	NA	16
Control LNA gapmer 4	G ^ A ^ ^m^C ^ c ^ a ^ a ^ t ^ c ^ t ^ c ^ g ^ t ^ t ^ A ^ G ^ T	NA	16

N, LNA; n, DNA; m, C5-methylcytosine; ^, phosphodiester; NA, not applicable.

**Table 2 ijms-24-14846-t002:** Analysis of sequence-dependent off-target binding of JEV RNA-targeted LNA gapmers to human sequences using GGGenome.

	Number of Regions Complementary to LNA Gapmers			
	0 MM or Gap	1 MM or Gap	2 MMs or Gaps	3 MMs or Gaps
Human genome				
LNA gapmers 7 and 8	0	1	77	4491
LNA gapmer 9	0	4	287	15,069
Ctrl LNA gapmer 1	0	31	1626	44,191
Ctrl LNA gapmer 2	0	1	315	15,601
Human RNA				
LNA gapmers 7 and 8	0	0	10	1673
LNA gapmer 9	0	0	81	5081
Ctrl LNA gapmer 1	0	0	178	8291
Ctrl LNA gapmer 2	0	0	59	3585

The numbers indicate human genome or RNA sequences complementary to the LNA gapmers with perfect matches or one, two, or three bases of mismatches (MMs) or gaps.

**Table 3 ijms-24-14846-t003:** Primer sequences and applications.

Primer Name	Sequence (5′→3′)	Position	Purpose
JaGAr 01_F	CCTGGGAATAGACTGGGAGAT	10,877–10,897	RT-qPCR
JaGAr 01_R	GTTCTTACTCACCACCAGCTACA	10,946–10,968	
GAPDH_F	GCCAGCCGAGCCACAT	NA	RT-qPCR
GAPDH_R	CTTTACCAGAGTTAAAAGCAGCCC	NA	
JE955f	TGYTGGTCGCTCCGGCYTA	955–973	Envelope protein
JE2536r	AAGATGCCACTTCCACAYCTC	2516–2536	
JE_LNA_1F	TCCAGGAAGACAGGGTCATC	10,372–10,391	LNA gapmer target region
JE_LNA_1R	CCTGTGTTCTTACTCACCACCAG	10,951–10,973	

Positions are based on JEV JaGAr 01 strain genome. Y = C or T.

## Data Availability

The data used and/or analyzed during the current study are available from the corresponding author on reasonable request.

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
