# Peer review of "Antiviral Efficacy of RNase H-Dependent Gapmer Antisense Oligonucleotides against Japanese Encephalitis Virus"

_ijms, 2023, doi:10.3390/ijms241914846_

Round 1

Reviewer 1 Report

In this paper Okamoto et al describe the design and efficacy of anti-sense oligonucleotides targeting the 3’UTR region of Japanese encephalitis virus as a potential therapeutic. The study demonstrate antiviral effectiveness of their candidate LNA gapmers in Vero and neuronal cells and further demonstrates that this antiviral activity is mediate via RNAse H dependent degradation of JEV RNA. The LNA gapmers also do not show any cytotoxicity and have a high degree of conservation across several genotypes of JEV.

There are a few comments regarding this study. Firstly, has the authors looked at whether the administration of their LNA Gapmers induce antiviral signalling within cells? It is logical to assume DNA sensors within the cell would detect the foreign DNA and this would lead to downstream antiviral signalling? This may potentially explain the higher effectiveness of LNA gapmer 8 in neuronal cells? This is definitely worth looking into and a caveat of the study. The study would also be strengthened from looking at effectiveness of the gapmers against the different genotypes of JEV beyond in silico assessment.

I would further suggest changing the viral titre graphs to a log scale. The non linear axes, while useful to show the difference between treatment and mock group, can tend to overstate minor differences. In supplementary figures S2 and S3c,d and e it would be helpful to have marker as well. Finally, in Table 2, LNA gaper 7 and 8 have one human RNA with one MM or gap and LNA gapmer 9 has 4 human RNA with one MM or gap. Which human genes were these RNA from, and what species of RNA was it? Can the authors identify the RNA and hypothesize as to what the potential impact of this interaction might be?

Reviewer 2 Report

Shunsuke Okamoto and coauthors discusses the antiviral efficacy of RNase H-dependent gapmer antisense oligonucleotides against Japanese encephalitis virus (JEV). The study found that LNA gapmers were effective in inhibiting JEV replication in vitro. The authors also analyzed the conservation of JEV genotypes and found that the LNA gapmers could potentially be applicable to different strains. This is an interesting study however, there are a few shortcomings in the manuscript and a few additional works are required.

For Inhibitory effect of LNA gapmers on JEV proliferation in Vero cells, authors should determine the cytotoxicity of LNA gapmers (0.05 - >5 µM) on Vero cells first. In figure 2 authors should show the cytotoxicity data first followed by inhibitory assay.

It is not clear why authors did not include LNA 4 for further validation (which showed more inhibition than LNA 7)?

Figure 4 and 5 should be merged. Cytotoxicity data on SK-N-SH cells should come before the antiviral assay.

In methodology section cytotoxicity assays (both Vero and SK-N-SH cells) should come before antiviral assay.

More experiments are needed to validate the antiviral activity of LNA gapmers at RNA and protein levels using qPCR and western blot respectively.

I strongly encourage the authors to include a schematic showing JEV target of LNA gapmer based therapy.

Round 2

Reviewer 1 Report

The authors have addessed all comment appropriately.